# Neither exogenous, nor endogenous: Evidence for a distinct role of negative emotion during attentional control

Xiaojuan Xue *, Gilles Pourtois

Cognitive & Affective Psychophysiology Laboratory, Department of Experimental Clinical & Health Psychology, Ghent University, Ghent, Belgium

* xiaojuan.xue@UGent.be

## Abstract

Negative or threatening stimuli capture attention. However, it remains unclear whether this phenomenon is best conceived as bottom-up (i.e., salience-driven) or top-down (i.e., goal-directed) instead. To address this question, we conducted two experiments using a previously validated dot-probe task (DPT) where physical salience (i.e., abrupt luminance change) and negative emotion (i.e., fearful face) competed with one another for attention selection (Experiment 1, n = 40) or negative (but also positive) emotion could be used as an endogenous cue by the participants to guide this process (Experiment 2, n = 39). Eye-tracking was used to ascertain that both cue and target were processed with peripheral vision. In Experiment 1, we found that negative emotion and physical salience both drove spatial attention in a bottom-up manner, yet their effects were under-additive, suggesting that they could mutually inhibit each other. Moreover, the results of Experiment 2 showed that fear, unlike happiness, could bias spatial attention in a top-down manner, but only when participants were aware of the association created between the emotional cue and target's location, leading to an enhanced validity effect in the high probability condition but an invalidity effect in the low probability one. Combined together, these novel findings suggest that negative value does not influence the priority map independently from physical salience and goal but depending on the specific combination of cues available for attention selection in the environment, it acts either as an exogenous or endogenous cue, thereby revealing an enhanced flexibility for it.

## Introduction

We are surrounded by an overwhelming amount of information. However, due to inherent processing capacity limitations at the cognitive level, only a restricted amount of it can be processed and eventually reach conscious awareness [1]. To cope with these limitations and enable an efficient selection of information, the brain is equipped

**Data availability statement:** All relevant data are within the manuscript and its Supporting Information files.

**Funding:** This work was supported by a grant (JY202126) from the Guangzhou Elite Scholarship Council (GESC) awarded to Xiaojuan Xue.

**Competing interests:** The authors have declared that no competing interests exist.

with selective attention that influences and organizes streams of sensory processing [2,3]. According to dominant models of attentional control in psychology and neuroscience, this selection is mostly determined either by bottom-up (stimulus-driven) or top-down (goal-directed) factors [4–7]. Bottom-up factors pertain to physical salience, where attentional control operates based on changes along low-level physical properties, such as color, size, motion, luminance, contrast, or abrupt onset/offset. These changes, especially if they are unexpected, capture attention and are prioritized during information processing [8–10]. Because this form of attentional control is mostly "external" and stimulus-driven, it is also referred to as exogenous attention in the literature [11]. However, most of the time, attentional control is not determined by salience or novelty alone, but instead by goals or instructions, which are "internalized" (or stored in memory) and subsequently used to afford selection of the relevant information available in the environment [6,12–15]. This form of attentional control corresponds to endogenous attention. According to a dominant framework [7], although these two forms of attentional control are dissociable, they can interact with one another, and exogenous attention can act as a "circuit breaker" when endogenous attention is used to prioritize the selection of novel or unexpected events. Recently, this model has been revised and improved to accommodate the effects of emotion (and motivation) on attentional control since they could hardly be explained either by salience or goal alone [16,17]. In this framework, "value" is assumed to provide a third independent pillar or component of the priority map, besides salience and goal, able to influence attentional control as well [16,18,19]. Value can be either positive/reward-related or negative/threat-related, and numerous behavioral studies have previously found that stimuli endowed with it capture attention in a way that is neither purely bottom-up nor top-down [20,21]. However, two unanswered questions are how attentional control is achieved (1) when salience and value directly compete with one another for selection, and (2) when value is an integral part or component of the goal. The goal of our study was to address these two open questions.

With respect to the first question, according to some models [22,23], effects of (negative) emotion (such as threat) on attention share similarities with exogenous attention and hence value might actually correspond to an instance of salience (see also Wolfe & Horowitz [8] for a review and discussion of this topic). For example, fear (conveyed by a fearful face) can enhance contrast sensitivity [24] and this perceptual effect appears to be mostly mediated by the processing of coarse/low-spatial frequency information [25]. However, happy faces, for which the upward-curved mouth provides an important diagnostic (low-level) feature, can also facilitate visual processing and bias spatial attention compared to neutral faces [26,27]. Hence, although threat-related stimuli can be prioritized during attentional selection (as revealed by faster/better processing for them compared to neutral or even positive stimuli in various tasks and contexts [28–30], including visual search [29,31], attentional blink [32,33], cueing [24,34], or dot-probe task (DPT) [35,36]), probably due to their enhanced evolutionary relevance [28–30], positive stimuli can also bias spatial attention under specific circumstances. Threat-related stimuli stand out perceptually and can lead to pop-out effects during visual search and/or enhanced spatial

orienting to their location, owing to their emotional significance. In this framework, emotional significance is determined by low-level physical properties and/or some diagnostic visual features [37,38]. Accordingly, V-shaped eyebrows [39–41], upward-shaped mouths [31], and wide eyes [42] capture attention because they manifest a potential threat (either anger or fear). Hence, negative value can "highjack" attentional control and this phenomenon is likely adaptive because the quick identification of potential threats or dangers in the environment can increase survival [21,43,44]. Likewise, (neutral) attributes such as luminance, size, color, motion, and contrast grab attention as well because they convey important information about a salient change in the environment that likely requires an appropriate behavioral response or action. In this framework, (negative) value could therefore correspond to another instance of salience implying that it should facilitate exogenous attention when it is driven by another low-level physical property (e.g., luminance). To test this hypothesis, we devised Experiment 1 where we provided participants with two competing bottom-up cues during a dot-probe task [30,35,36,45]. Exogenous attention was fostered by the use of a low-level physical change at the cue level, namely luminance [4,8,46]. Orthogonally to exogenous attention, the cue had either a threat-related or neutral value, as achieved by the use of fearful and neutral faces, respectively [30,47]. As a control condition, we used happy faces. In this condition, the same abrupt luminance change occurred on one side of the display, as was done with fearful faces, eventually yielding a comparable exogenous cueing for these two emotion categories. We reasoned that if negative value is just another instance of salience, then additive effects of luminance and threat on spatial attention should be observed whereby target processing should benefit (and be the most efficient) from the combination of both cues [48].

Regarding the second interrogation and in stark contrast with the exogenous attention studies reviewed here above, other ones have shown that negative value captures attention when it is an integral part or component of the goal, implying that value is actually close to goal and hence top-down attention [49–51]. These findings accord with recent theoretical models assuming that emotion processing necessarily depends on dedicated goal-directed processes and is not automatic [52]. For example, Vogt et al. [49] used a DPT in combination with induction trials meant to activate a specific goal and found that negative stimuli only captured attention when they were directly task-relevant and hence goal-relevant [45,49,53–55]. More specifically, during the induction trials, the participants were asked to press a key whenever a specific and pre-defined (emotional) stimulus was presented but to withhold responding to other stimuli, making this former stimulus task and hence goal relevant. These studies suggest that negative emotion can capture attention but this effect results from top-down attention control mechanisms, suggesting close ties between value and goal [56]. Moreover, as we have recently shown in a series of behavioral experiments based on the DPT [45], the modulation of emotional attention by means of induction trials appears to depend on the contingency between value and goal. More specifically, in this recent study, we found that negative emotion ceased to capture attention if and only if a high contingency was artificially created between goal and value, suggesting some important boundary conditions for this modulation. Moreover, in many spatial attention studies with neutral stimuli performed in the past, goal was not manipulated by means of separate induction trials, but instead cue-target probability, where (symbolic) cues informed participants about target location with a certain probability [15,57]. This manipulation turned out to be powerful enough to yield an endogenous control of attention. Here we sought to borrow this logic and assess in Experiment 2 whether (negative) value could foster an endogenous control of attention or not (see also [58]). To this end, in Experiment 2, we also used a DPT, but value was not used as a distractor (as in Experiment 1), but instead, information for the participants to predict target's location. More specifically, participants were informed beforehand that the target was mostly shown on the side occupied by the neutral face at the cue level in some blocks (low probability condition) or the emotional face in other ones (high probability condition). Hence, low vs. High probability simply refers to the likelihood (either low or high) that the target appeared on the same side as the emotional face at the cue level. Accordingly, in the low probability condition, the target appeared on 75% of the trials on the same side as the neutral face, while it was shown on 25% of the trials on the same side as the emotional face instead. In comparison, in the high probability condition, this mapping was reversed: it appeared on 75% of the trials on the same side as the emotional face, whereas on 25% of the trials, it was shown on the same side as the neutral face. As in Experiment 1, this emotional face could be

either fearful or happy. We hypothesized that if the capture of attention by negative value depends on goal, then the spatial orienting to negative emotion should be larger in the high than low probability condition.

In both experiments, we used and adapted a DPT along with eye-tracking that we have validated recently [45]. In this previous study, we compared attentional biases to emotional faces using a DPT alone where emotion was never goal-relevant (Experiment 1) or made directly task-relevant by means of induction trials (Experiments 2–3). We found out that, in the absence of induction trials (Experiment 1), negative faces captured attention, with faster target processing when it appeared on the same side as the preceding fearful face (i.e., fear-valid trials) compared to the opposite side where the neutral face was shown (i.e., fear-invalid trials), but also when it appeared on the side of the preceding neutral face (i.e., happy-invalid trials) compared to the happy face (i.e., happy-valid trials), suggesting that negative emotion, as opposed to fear or threat per se, actually captured attention in this task. Accordingly, in Experiment 1 of the current study, we assessed if the same behavioral results could be found when besides value (i.e., actual emotional expression of the face used at the cue level), a low-level visual change based on luminance occurred and hence yielded an exogenous control of attention. In Experiment 2, we used the same DPT again, but altered cue-target probability to examine whether (negative) value could guide and influence endogenous attention or not.

## Materials and methods

### Participants

A comparable sample size to our previous study [45] was used, with a total of 83 participants recruited: 40 for Experiment 1 and 43 for Experiment 2. The data of one participant in Experiment 1 were removed due to poor accuracy (ACC) (i.e., it fell below 3 SDs below the mean). In Experiment 2, the data of one participant were removed because of eye-tracking problems, and of another one because of poor ACC (same criterion as Experiment 1). Hence, the data of 80 participants were retained for further analyses (Experiment 1: 39 participants, aged 18−35, mean age = 23.90 years, SD = 4.18 years, 7 males; Experiment 2: 41 participants, aged 18−27, mean age = 21.49 years, SD = 1.94 years, 10 males). These participants were recruited online using Sona, which is administered by Gent University. All participants were right-handed and had normal or corrected-to-normal vision, no history of neurological or psychological impairment, and no current medication. They provided written informed consent and were compensated 10 euros for their participation. The study was approved by the local ethical committee of the faculty of psychology and educational sciences at Ghent University (file number: #2022−029).

At the end of the experiment, participants were asked to complete the trait version of the State-Trait Anxiety Inventory (STAI-T; see Spielberger [59]). We used the twenty-item version (Form Y-2) that assessed trait anxiety. All items were rated on a 4-point scale (e.g., from "Almost Never" to "Almost Always"). Scores range from 20 to 80, with higher scores reflecting higher levels of anxiety. We administered the STAI to our participants because anxiety, as measured using this specific scale, has been shown to influence emotional attention in some previous studies [60,61]. Moreover, we also used it to assess if the anxiety levels of our participants were eventually comparable between the two experiments performed. The average score of the STAI was 43.80 (range 26−68; SD: 10.265) in Experiment 1 and 38.27 (range 29−63; SD: 8.155) in Experiment 2. The mean score in Experiment 2 was significantly lower than in Experiment 1 ($t_{78}$ = −2.673, $p$ = 0.009, Cohen's d = −0.598). Despite this significant difference, both scores were lower than the commonly used cutoff of 45 for clinical anxiety, indicating that neither group reached clinical anxiety [28,62]. Participants' data was collected between September 2023 and March 2024 and was anonymized at the time of collection.

### Apparatus and stimuli

Participants were seated approximately 70 cm from a 19-inch CRT screen with a resolution of 1024 × 768 pixels (60 Hz). Their head was stabilized using a chinrest in a soundproof experimental room. Stimulus presentation and response recording were managed using E-Prime software (Version 3.0), with responses collected via a response pad. The left eye position was continuously monitored using an Eyelink 1000 + eye-tracking system (SR Research) at a sampling rate of

1000 Hz. Synchronization between stimulus presentation and eye-tracking was achieved using specific E-Prime Extensions for Eyelink. A 9-point calibration procedure was used at the beginning as well as in the middle of the experiment.

The stimuli comprised emotional faces from the Ekman and Friesen dataset [63], featuring 10 distinct identities (5 male and 5 female). For each identity, fearful, happy, and neutral expressions were selected, yielding a total of 30 different face stimuli. Each face was trimmed to remove hair, ears, neck, and non-facial information, and was shaped into an oval measuring 6 × 8 cm. The images were converted to grayscale and adjusted in ImageJ. Non-parametric analysis using the Kruskal–Wallis test indicated no significant differences in mean luminance and contrast across the three emotion categories (luminance: $H(2) = 2.821$, $P = 0.244$; contrast: $H(2) = 1.506$, $P = 0.471$).

A total of 160 face pairs were constructed as cues. Each pair consisted of two different identities of the same gender, with one face displaying an emotional expression (fearful or happy) and the other a neutral one. Faces were positioned 8 cm from a central fixation cross (1 × 1 cm) along the horizontal axis, with one on the left and the other one on the right side. Four face pair combinations were created to balance the presentation of neutral and emotional faces on both sides: fearful-neutral, neutral-fearful, happy-neutral, and neutral-happy. Each combination included 40 different pairs. Additionally, in Experiment 1 only, two white rectangles surrounding each face of the pair were used at the cue level. We used these rectangles to yield a low-level visual change in one of them (i.e., luminance change, as achieved by increasing the thickness of this rectangle) that captured attention exogenously. These two frames, both with outer dimensions of 8 cm by 8.5 cm, differed in border width: one frame (i.e., the thick one) had a 10-pixel (2 mm) border, while the other one (i.e., the thin one) had a 2-pixel (0.4 mm) border. Luminance was measured using a calibrated photometer under controlled lighting conditions. For each rectangle, we subtracted the inner area (black) from the total outer area (white). The corresponding value of the rectangle was multiplied by 60 cd/m². Results showed that the thick rectangle (0.03864 cd) was almost five times more luminant than the thin one (0.00786 cd).

Following the cue, a unilateral target measuring 3 × 3 cm was presented. This target was a dark grey square (hex code #131313) that could be tilted 45 degrees to become a diamond. Thus, the target appeared either as a square or a diamond. On each side of the screen, two white placeholders (rectangles in Experiment 1 and double brackets in Experiment 2) were presented alongside the unilateral target to enhance target processing at the locations where the faces had been shown. These placeholders, sized 6 × 8 cm, matched the faces shown at the cue level. Given the uniform black background and the low-contrast grey target, participants needed to covertly orient spatial attention to discern the target's shape. Each target type (square or diamond) was presented with an equal probability in both experiments.

## Procedure

All stimuli were shown on a black background. Experiment 1 (see Fig 1A) consisted of one practice block of 20 trials, followed by 8 experimental blocks of 80 trials (amounting to 640 trials). Each trial began with a fixation cross shown for 500 ms, followed by the emotional cue (i.e., a pair of faces) shown for 100 ms either within the cued frame (i.e., a thick rectangle) or within the uncued one (i.e., a thin rectangle), with an equal probability on either the left or right side. After a short, variable, and equiprobable interval (100, 150, 200, 250, or 300 ms), the target was presented for 150 ms. We mostly used these different/variable SOAs to prevent temporal attention effects and in keeping with a previous study [45] where this variability was introduced (with the DPT). As a result of it, the participants could not easily anticipate the actual onset of the target. Participants were asked to discriminate the shape of the target, either a diamond or a square, as accurately and quickly as possible. Speed was emphasized via on-screen instructions shown before each block and it provided the main dependent variable of interest in both experiments. A maximum response window of 2s was imposed on every trial; if participants failed to respond within 2s, it was automatically terminated, and the task proceeded to the next trial. After the response (or 2000 ms in case of no response), a 500ms interval was used before the next trial started. Participants used their left index finger for the diamond and their right index finger for the square, with this mapping being alternated across them.

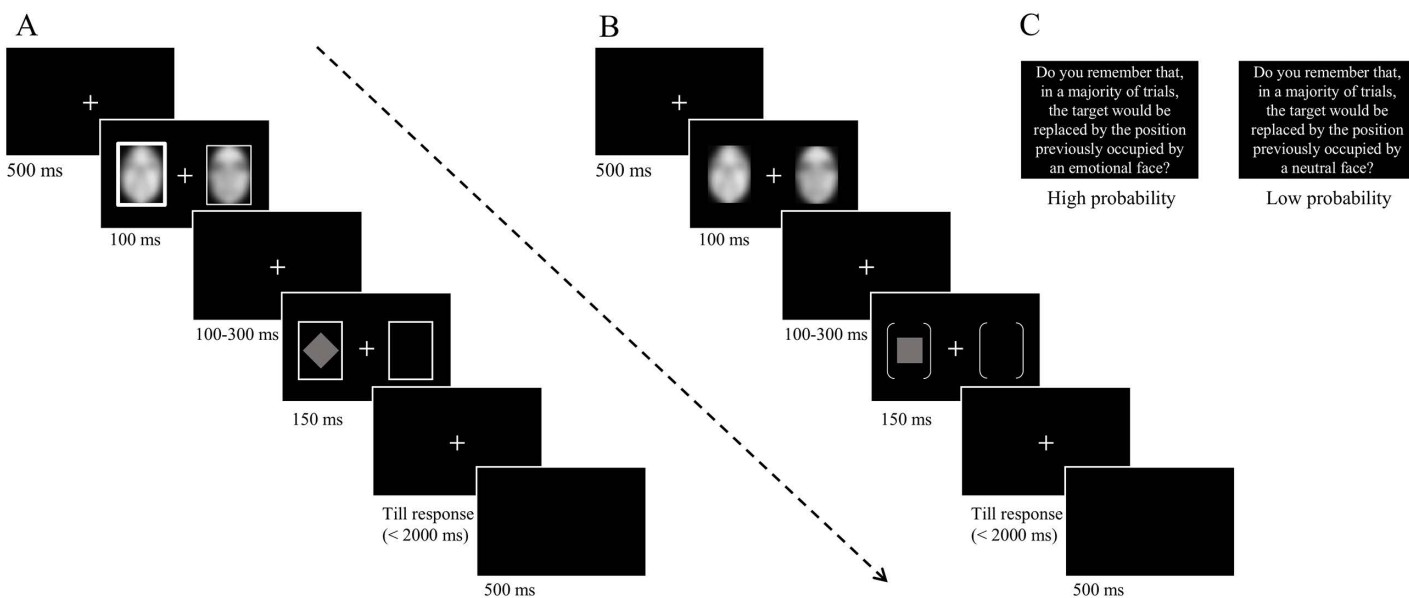

**Fig 1. Trials structure.** (A) Structure of a DPT trial in Experiment 1. In this example, the fearful face and cued rectangle (exogenous attention) are both presented on the left side, but these two validities were orthogonal. (B) Structure of a DPT trial in Experiment 2. (C) A screen was used at the beginning of each block in Experiment 2 to remind participants of the specific association created between cue and target during it. Note: To comply with copyright restrictions, the faces in this figure are shown in blurred form. In the actual experiments, participants were presented with the original, unblurred stimuli from Ekman & Friesen (1976).

Experiment 2 (see Fig 1B) was divided into two parts with different probabilities of valid and invalid trials (see here below). This manipulation was used to induce endogenous shifts of spatial attention. Moreover, in Experiment 2 (and unlike Experiment 1), luminance (and hence exogenous cueing) was not manipulated. Each session included one practice block of 20 trials, followed by 4 experimental blocks of 80 trials. After these 4 blocks, participants' awareness of the association between the cue and the target was probed using a visual analog scale (VAS). Using this scale, they were asked to indicate their explicit knowledge of the association created between the cue and the target. Hence, in the low probability condition, they were asked to indicate on a scale ranging from 0 (no knowledge) to 100 (full knowledge) how aware they were of the fact that the target most of the time appeared on the same side as the neutral face. Similarly, in the high probability condition, they were asked to rate their awareness of the contingency created between target location and side of the emotional face. Hence, higher scores on this VAS indicated greater awareness at the subjective level. At the beginning of each session, the participants received specific instructions (on screen) about the association created between cue and target (i.e., in these blocks, on a majority of trials, the target will appear at the location previously occupied by the neutral face in the pair or the emotional face, depending on the session). At the beginning of each block, they were reminded about this association via written instructions shown on screen (see Fig 1C). Each trial started with a 500 ms central fixation cross, followed by a cue (i.e., a pair of faces) presented for 100 ms. After a random, variable, and equiprobable SOA (100, 150, 200, 250, or 300 ms), the target (either a diamond or a square) was shown for 150 ms at the location previously occupied by the emotional face (valid trial) or the neutral face (invalid trial). Participants were asked to discriminate the shape of the target using their left index finger for the diamond and their right index finger for the square, with this mapping being alternated across them. In one session (i.e., low probability condition), the target appeared on the side previously occupied by the neutral face on 75% of the trials and on the side of the emotional face on 25% of the trials. In the other session (i.e., high probability

condition), this rule was reversed: the target appeared on the side previously occupied by the emotional face on 75% of the trials and on the side of the neutral face on 25% of the trials. At the end of each session, their awareness of this association was measured using the VAS. The order of these two conditions (or sessions) was counterbalanced between subjects.

## Data analysis

Data analyses were conducted with Matlab R2023a (The Mathworks Inc., Natick, MA, USA). To ensure that participants processed all stimuli with peripheral vision, we removed offline the trials where the eye deviated more than 3 degrees away from the central fixation cross (see Table 1–2 as well as S4–S7 Figs in S1_file; see also Xue & Pourtois [45] for a similar procedure). To this aim, we created a region of interest (ROI) around it, where the border was delimited by a circle (X values: 417.28–606.72 pixels; Y values: 289.28 to 478.72 pixels), whose diameter was 3.7 cm at a viewing distance of 70 cm.

For the trials that were not contaminated by eye movements (see Tables 1–2), ACC and RTs for correct responses were analyzed using JASP (version 0.17). Data visualization was carried out in R Studio (3.3.0), using the ggplot2 package. For ACC, for each participant separately, the first trial of each block and outliers (defined using a ±3 SDs criterion above/below the grand mean) were excluded. For the RT data, the first trial of each block, incorrect trials, and outliers were excluded from further analyses. Following this pre-processing, in Experiment 1, a three-way repeated-measures ANOVA with Emotion (either happy or fearful face at the cue level), Validity-V (here "V" refers to value of the face: the target replaced either the emotional or neutral face at the cue level, corresponding to valid and invalid trials, respectively), and Validity-E (here "E" refers to exogenous cue: the target replaced either the thick or thin rectangle at the cue level, corresponding to valid and invalid trials, respectively) as within-subject factors was used to analyze the RTs (as well as

**Table 1. The number of trials kept per condition in Experiment 1.**

| | | Emotion | | | | | |
| --- | --- | --- | --- | --- | --- | --- | --- |
| | | Fear | | | Happy | | |
| Cue | Validity | Total | Eye-tracking correction | Correct trials | Total | Eye-tracking correction | Correct trials |
| Cued | Valid | 80 | 72.97 (6.63) | 68.64 (6.39) | 80 | 73.26 (6.40) | 69.23 (7.41) |
| | Invalid | 80 | 73.00 (7.06) | 69.15 (7.51) | 80 | 72.85 (7.25) | 69.05 (7.91) |
| Uncued | Valid | 80 | 73.10 (7.13) | 68.59 (7.36) | 80 | 72.23 (7.95) | 67.46 (8.81) |
| | Invalid | 80 | 72.94 (6.94) | 69.36 (7.10) | 80 | 73.03 (6.59) | 68.97 (6.70) |

*Note*: Standard deviation is shown in parentheses. "Total" refers to the total number of trials presented per condition; "Eye-tracking correction" to the mean number of trials kept after exclusion of excessive eye-movements; "Correct trials" to mean number of trials kept with correct responses."

**Table 2. The number of trials kept per condition in Experiment 2.**

| | | Emotion | | | | | |
| --- | --- | --- | --- | --- | --- | --- | --- |
| | | Fear | | | Happy | | |
| Probability | Validity | Total | Eye-tracking correction | Correct trials | Total | Eye-tracking correction | Correct trials |
| Low | Valid | 40 | 37.38 (3.53) | 35.24 (3.62) | 120 | 38.00 (4.62) | 36.19 (4.51) |
| | Invalid | 120 | 114.10 (9.89) | 107.69 (9.56) | 40 | 113.12 (11.82) | 107.76 (48.00) |
| High | Valid | 120 | 112.86 (10.12) | 107.41 (10.39) | 40 | 113.52 (10.44) | 107.10 (10.73) |
| | Invalid | 40 | 38.26 (3.37) | 36.12 (3.63) | 120 | 37.43 (2.79) | 35.60 (3.01) |

*Note*: Standard deviation is shown in parentheses. "Total" refers to the total number of trials presented per condition; "Eye-tracking correction" to the mean number of trials kept after exclusion of excessive eye-movements; "Correct trials" to mean number of trials kept with correct responses."

ACC, which is reported in the S1_file, see S1 Fig). In Experiment 2, a three-way repeated-measures ANOVA with Emotion, Validity (same as Validity-V in Experiment 1), and Probability (either low or high probability that the target appeared at the location of the emotional face) as within-subject factors were used to analyze the RTs (see S2 Fig for ACC; S1_file). We reported partial eta square ($\eta_p^2$) values as an estimate of effect size. A Bonferroni correction was used for post-hoc comparisons. For all these analyses, the significance level was set to $p < 0.05$, and Bayes factors (BF) were also calculated using JASP (version 0.17), enabling us to quantify the amount of evidence gathered in favor of the null ($H_0$) or the alternative hypothesis ($H_1$).

For completeness, we also computed attentional bias scores (ABSes) by subtracting invalid from valid trials for ACC, and valid from invalid trials for RTs. A positive score indicates (enhanced) orienting to the emotional face while a negative score indicates (enhanced) orienting to the neutral face in the pair. Moreover, in Experiment 2, we conducted Pearson's correlations between these ABSes (either fear or happiness) and awareness of the cue-target association (ABSes based on ACC is reported in the S1_file, see S3 Fig). In control analyses, we also partialled out anxiety (as measured by the STAI). In order to directly compare the role of fear vs. happiness in awareness at the statistical level, we constructed a generalized linear model (GLM) that included both fear (F_ABS) and happiness (H_ABS) as concurrent predictors of it (i.e., Scale = $\beta_0 + \beta_1 \times$ F_ABSes $+ \beta_2 \times$ H_ABSes $+ \varepsilon$).

## Results

### Experiment 1

For the RTs (see Fig 2A), the ANOVA showed a significant main effect of Emotion ($F_{1,38} = 23.635$, $p < 0.001$, $\eta_p^2 = 0.383$; $BF_{incl} = 441.720$ suggesting decisive evidence for including this effect), with faster RTs for trials where a happy face was used at the cue level compared to a fearful face. The main effect of Validity-V was also significant ($F_{1,38} = 5.230$, $p = 0.028$, $\eta_p^2 = 0.121$; $BF_{incl} = 1.271$ suggesting anecdotal evidence for including this effect), showing faster RTs for Invalid than Valid trials (see Fig 2B). The main effect of Validity-E was significant as well ($F_{1,38} = 18.816$, $p < 0.001$, $\eta_p^2 = 0.331$; $BF_{incl} = 85.362$ suggesting very strong evidence for including this effect), indicating faster RTs for valid than invalid trials (see Fig 2B). Moreover, the interaction between Emotion and Validity-V was significant ($F_{1,38} = 12.815$, $p < 0.001$, $\eta_p^2 = 0.252$; $BF_{incl} = 31.486$ suggesting very strong evidence for including this effect). Post-hoc tests showed faster RTs for happy-invalid than happy-valid ($t_{38} = -4.178$, $p < 0.001$, Cohen's $d = -0.115$), fear-valid ($t_{38} = -5.034$, $p < 0.001$, Cohen's $d = -0.132$), and fear-invalid trials ($t_{38} = -5.920$, $p < 0.001$, Cohen's $d = -0.161$). The difference between fear-valid and fear-invalid trials was not significant ($t_{38} = -1.060$, $p = 0.585$, Cohen's $d = -0.029$). The interaction between Validity-V and Validity-E was also significant ($F_{1,38} = 7.563$, $p = 0.009$, $\eta_p^2 = 0.166$; $BF_{incl} = 1.962$ suggesting anecdotal evidence for including this effect). Post-hoc tests showed faster RTs for valid than invalid trials based on the exogenous cue, yet only when they were also valid in terms of value ($t_{38} = -5.092$, $p < 0.001$, Cohen's $d = -0.116$). In comparison, this difference was not significant when value was invalid ($t_{38} = -1.569$, $p = 0.363$, Cohen's $d = -0.036$) (see Fig 2C). Neither the two-way interaction between Emotion and Validity-E ($F_{1,38} = 0.022$, $p = 0.884$, $\eta_p^2 < 0.001$) nor the three-way interaction between Emotion, Validity-V, and Validity-E was significant ($F_{1,38} = 0.469$, $p = 0.497$, $\eta_p^2 = 0.012$). A Bayesian ANOVA further showed that the model including Emotion + Validity-V + Validity-E + Emotion x Validity-V + Validity-V x Validity-E best explained these data (see Table 3).

Because we used variable intervals/SOAs between the cue and the target (i.e., 100 ms, 150 ms, 200 ms, 250 ms, and 300 ms), we ran a control analysis with "SOA" included as a fourth within-subject factor (see Fig 3). Given the limited number of trials per cell/SOA, we grouped the 100 ms and 150 ms SOAs into a short category, while the 250 ms and 300 ms SOAs were lumped together in a long SOA category. In this control analysis, we therefore excluded the 200 ms SOA condition (see Table 4). For the RTs, the ANOVA showed that the main effects of Emotion ($F_{1,38} = 23.98$, $p < 0.001$, $\eta_p^2 = 0.387$), Validity-V ($F_{1,38} = 10.00$, $p = 0.003$, $\eta_p^2 = 0.208$), and Validity-E ($F_{1,38} = 7.75$, $p = 0.008$, $\eta_p^2 = 0.169$) were significant. The main effect of SOA was significant as well ($F_{1,38} = 12.68$, $p = 0.001$, $\eta_p^2 = 0.250$), with faster RTs for the long

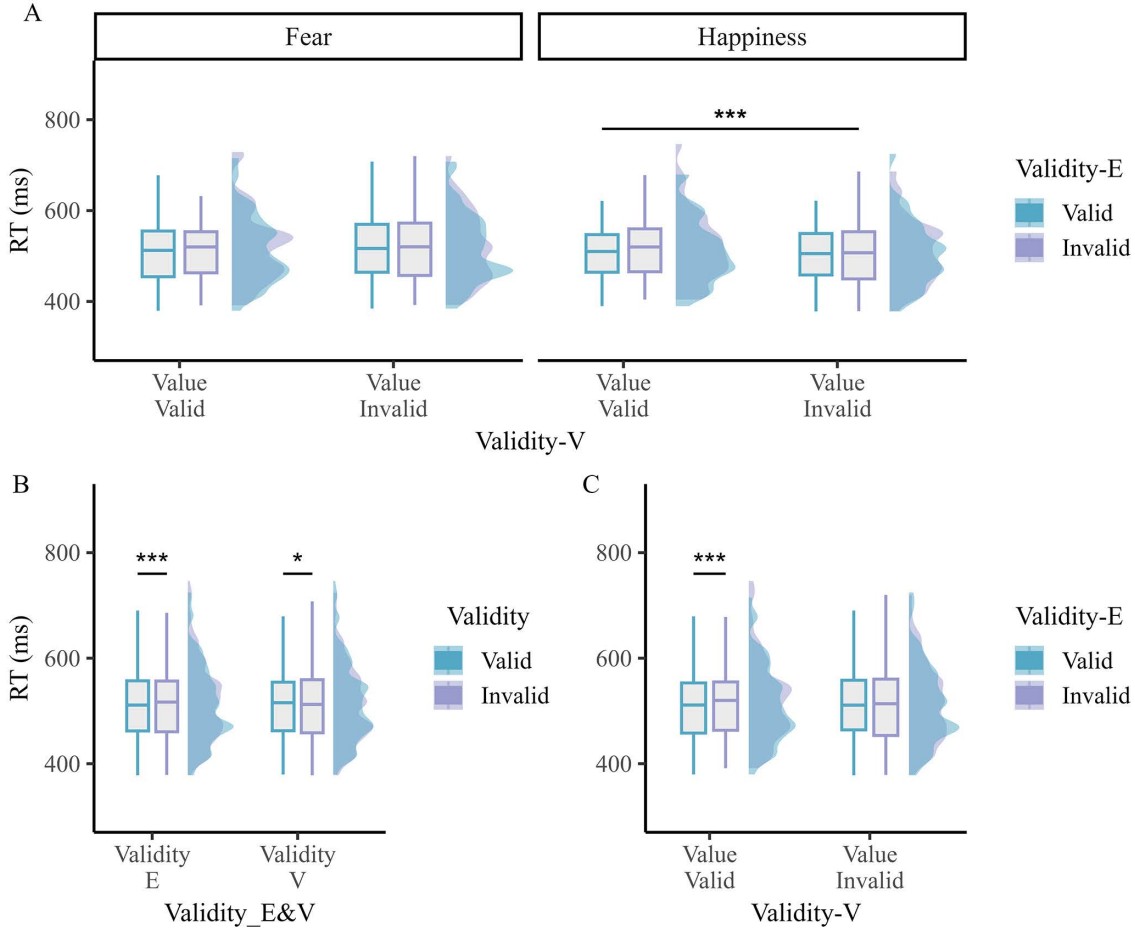

**Fig 2. RTs results in Experiment 1.** (A) RTs for each condition separately. (B) Significant main effects of Validity-E and Validity-V. (C) Significant interaction between Validity-E and Validity-V. In each graph, the mean performance is shown in a boxplot along with its distribution (half-density with color). * $p < 0.05$, *** $p < 0.001$.

than the short SOA. Moreover, the interaction between Emotion and Validity-V was significant ($F_{1,38} = 11.34$, $p = 0.001$, $\eta_p^2 = 0.230$). Post-hoc tests showed faster RTs for happy-invalid than happy-valid ($t_{38} = -4.619$, $p < 0.001$, Cohen's $d = -0.130$), fear-valid ($t_{38} = -5.690$, $p < 0.001$, Cohen's $d = -0.157$), and fear-invalid trials ($t_{38} = -5.817$, $p < 0.001$, Cohen's $d = -0.163$). The difference between fear-valid and fear-invalid trials was not significant ($t_{38} = -0.218$, $p = 0.828$). The interaction between Validity-V and Validity-E was also significant ($F_{1,38} = 7.22$, $p = 0.011$, $\eta_p^2 = 0.160$). Post-hoc tests showed faster RTs for valid than invalid trials based on the exogenous cue, yet only when they were also valid in terms of value ($t_{38} = -3.862$, $p < 0.001$, Cohen's $d = -0.092$). In comparison, this difference was not significant when value was invalid ($t_{38} = -0.370$, $p = 1.000$). All other effects remained non-significant: two-way interaction between Emotion and Validity-E ($F < 0.001$, $p = 0.993$), between Emotion and SOA ($F = 0.173$, $p = 0.680$), between Validity-V and SOA ($F = 1.123$, $p = 0.296$), between Validity-E and SOA ($F = 0.007$, $p = 0.935$), the three-way interaction between Emotion, Validity-V and Validity-E ($F = 0.010$, $p = 0.923$), between Emotion, Validity-V and SOA ($F = 0.644$, $p = 0.427$), between Emotion, Validity-E and SOA ($F = 3.810$, $p = 0.058$), between Validity-V, Validity-E and SOA ($F = 0.148$, $p = 0.702$), and the four-way interaction between Emotion, Validity-V, Validity-E and SOA ($F = 0.004$, $p = 0.952$)

**Table 3. Bayesian model comparison in Experiment 1. All models were compared with the best model. Other models (BF$_{10}$ =<0.01) are not shown.**

| Models | P(M) | P(M\|data) | BF$_M$ | BF$_{10}$ | error % |
|---|---|---|---|---|---|
| Emotion + Validity-V + Validity-E + Emotion x Validity-V + Validity-V x Validity-E | 0.053 | 0.504 | 18.259 | 1.000 | |
| Emotion + Validity-V + Validity-E + Emotion x Validity-V | 0.053 | 0.258 | 6.267 | 0.513 | 4.950 |
| Emotion + Validity-V + Validity-E + Emotion x Validity-V + Emotion x Validity-E + Validity-V x Validity-E | 0.053 | 0.103 | 2.077 | 0.205 | 3.954 |
| Emotion + Validity-V + Validity-E + Emotion x Validity-V + Emotion x Validity-E | 0.053 | 0.052 | 0.985 | 0.103 | 4.095 |
| Emotion + Validity-V + Validity-E + Emotion x Validity-V + Emotion x Validity-E + Validity-V x Validity-E + Emotion x Validity-V x Validity-E | 0.053 | 0.043 | 0.811 | 0.086 | 11.518 |
| Emotion + Validity-V + Validity-E + Validity-V x Validity-E | 0.053 | 0.016 | 0.301 | 0.033 | 4.184 |
| Emotion + Validity-V + Validity-E | 0.053 | 0.008 | 0.140 | 0.015 | 3.659 |
| Emotion + Validity-E | 0.053 | 0.006 | 0.110 | 0.012 | 3.314 |

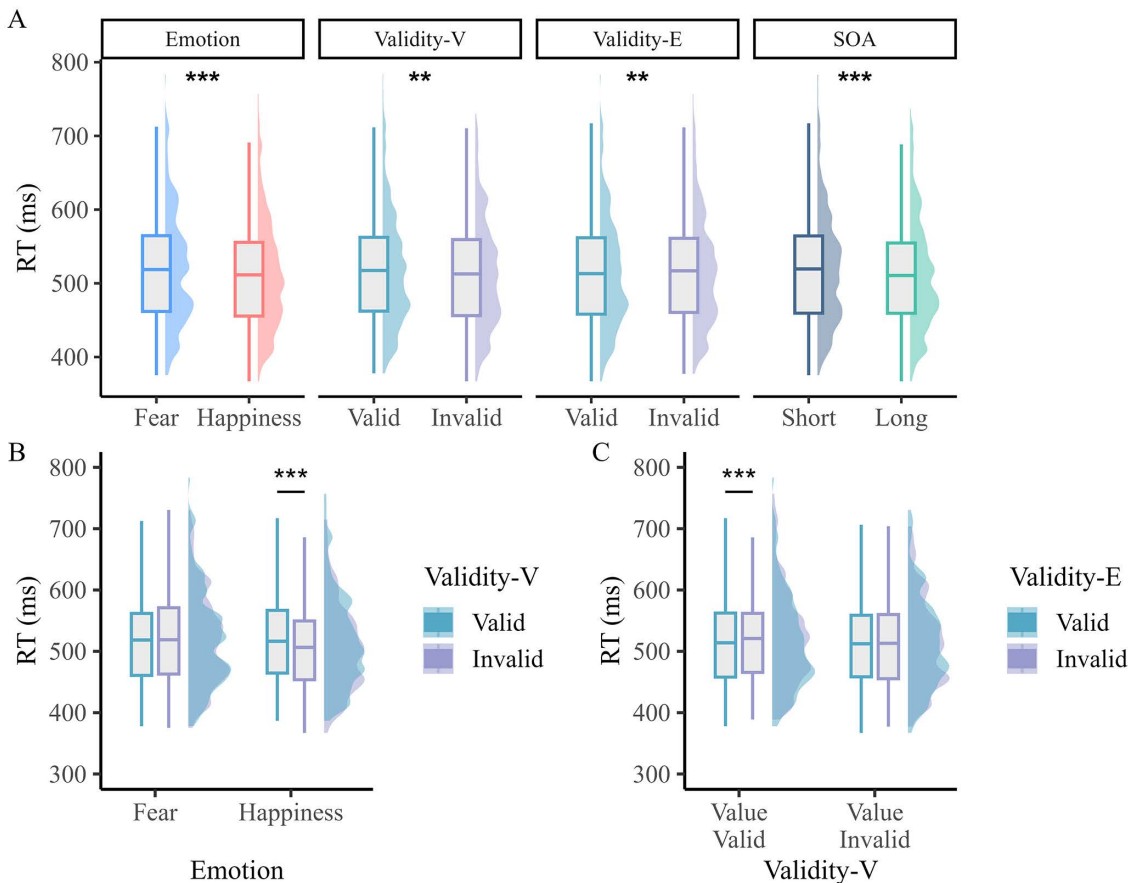

**Fig 3. Effect of SOA in Experiment 1.** (A) Significant main effects of Emotion, Validity-V, Validity-E, and SOA. (C) Significant interaction between Emotion and Validity-V. (C) Significant interaction between Validity-E and Validity-V. In each graph, the mean performance is shown in a boxplot along with their distribution (half-density with color). ** $p < 0.01$, *** $p < 0.001$.

**Table 4. The number of trials kept per condition in Experiment 1 when SOA was considered.**

| Cue | Validity | SOA | Emotion Fear Eye-tracking correction | Correct trials | Happy Eye-tracking correction | Correct trials |
|---|---|---|---|---|---|---|
| Cued | Valid | S | 28.13 (3.53) | 26.43 (3.30) | 30.80 (2.90) | 29.15 (3.56) |
| | | L | 29.00 (2.25) | 27.05 (2.64) | 27.62 (2.89) | 26.23 (3.15) |
| | Invalid | S | 24.23 (2.41) | 24.20 (2.39) | 30.56 (4.00) | 30.45 (3.98) |
| | | L | 29.28 (3.32) | 27.59 (3.57) | 27.51 (2.97) | 26.39 (3.34) |
| Uncued | Valid | S | 30.80 (3.08) | 28.73 (3.30) | 27.64 (2.79) | 25.70 (3.42) |
| | | L | 29.31 (3.48) | 27.51 (3.45) | 30.10 (2.61) | 28.23 (3.88) |
| | Invalid | S | 31.46 (3.73) | 31.47 (3.72) | 25.51 (2.74) | 25.48 (2.72) |
| | | L | 28.31 (3.56) | 26.82 (3.55) | 30.21 (3.24) | 28.72 (3.38) |

*Note*: Standard deviation is shown in parentheses. S: short SOA; L: long SOA.

## Experiment 2

For the RTs (see Fig 4), the ANOVA only showed a significant interaction between Emotion, Validity, and Probability ($F_{1,40}$=8.317, $p$=0.006, $\eta_p^2$=0.172; $BF_{incl}$=5.423 suggesting moderate evidence for including this effect), but none of the post hoc tests were significant. The main effects of Emotion, Validity, and Probability were all non-significant (all $Fs$<1, $ps$>0.5). Moreover, the interaction between Emotion and Validity, Emotion and Probability, and Validity and Probability all were non-significant (all $Fs$<1, $ps$>0.5). A Bayesian ANOVA showed that the null model and the model with probability comparably explained the data best (see Table 5). When exploring the significant three-way interaction further, the results showed that in the low probability condition, the interaction between Emotion and Validity was significant ($F_{1,40}$=4.052, $p$=0.051, $\eta_p^2$=0.092; $BF_{incl}$=1.822 suggesting anecdotal evidence for including this effect). However, none of the post-hoc tests reached significance. In comparison, in the high probability condition, the interaction between emotion and validity was not significant ($F_{1,40}$=3.056, $p$=0.088, $\eta_p^2$=0.071; $BF_{incl}$=1.293 suggesting anecdotal evidence for including this effect). Similarly, none of the post-hoc tests reached significance.

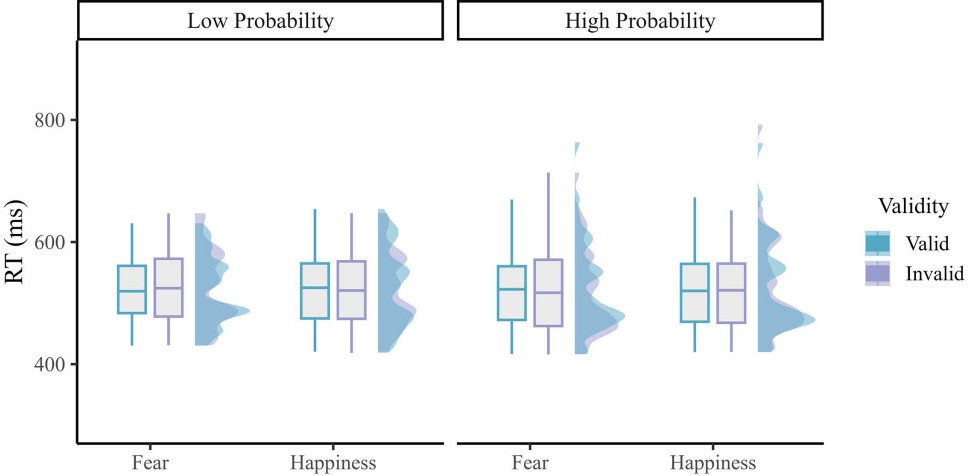

**Fig 4. RTs results in Experiment 2 for the low and high probability conditions, separately.** In each graph, the mean performance is shown in a boxplot along with their distribution (half-density with color).

**Table 5. Bayesian Model Comparison in Experiment 2. All models were compared with the best model. Other models (BF$_{10}$ = <0.01) are not shown.**

| Models | P(M) | P(M\|data) | BF$_M$ | BF$_{10}$ | error % |
|---|---|---|---|---|---|
| Probability | 0.053 | 0.786 | 66.272 | 1.000 | |
| Null model (incl. subject and random slopes) | 0.053 | 0.148 | 3.132 | 0.188 | 99.948 |
| Validity | 0.053 | 0.029 | 0.532 | 0.037 | 99.969 |
| Emotion | 0.053 | 0.028 | 0.526 | 0.036 | 100.154 |

Regarding awareness, the mean score was 65.37 in the low probability condition (range 50−95; SD: 9.797) and 65.488 in the high probability condition (range 25−90; SD: 10.812), respectively. They were both significantly higher ($t_{40}$ = 10.042, $p < 0.001$, Cohen's $d$ = 1.568, $t_{40}$ = 9.172, $p < 0.001$, Cohen's $d$ = 1.432, respectively) than 50 (i.e., chance level). They were not statistically different from each other ($t_{40}$ = 0.079, $p = 0.938$, Cohen's $d$ = 0.012), but were significantly positively correlated with each other ($r = 0.542$, $p < 0.001$). For the ABSes calculated using RTs, we found that awareness correlated negatively with fear ($r = −0.480$, $p = 0.001$) in the low probability condition (see Fig 5A). However, no correlation was found between awareness and happiness in this condition ($r = 0.041$, $p = 0.797$). Interestingly, in the high probability condition, awareness was positively correlated with fear ($r = 0.351$, $p = 0.025$) (see Fig 5B), while again, no such correlation was found with happiness ($r = −0.046$, $p = 0.776$). Moreover, when controlling for levels of anxiety (using the STAI), this intriguing dissociation between fear and happiness remained, in both conditions (low probability: fear: $r = −0.473$, $p = 0.002$; happiness: $r = 0.018$, $p = 0.911$; high probability: fear: $r = 0.382$, $p = 0.015$; happiness: $r = −0.030$, $p = 0.856$). Importantly, The GLM results (see Table 6) confirmed that in the low probability condition, fear but not happiness was a significant predictor of awareness: F_ABS (attentional bias score for fear) ($t_{40}$ = −3.409, $p = 0.002$) and H_ABS (attentional bias score for happiness) ($t_{40}$ = 0.643, $p = 0.524$). Similarly, in the high probability condition, awareness was selectively predicted by fear ($t_{40}$ = 2.299, $p = 0.027$) but not happiness ($t_{40}$ = −0.212, $p = 0.833$).

## Discussion

Previous studies have suggested that attentional control does not operate solely in a bottom-up or top-down manner, but that value provides a third additional source of information able to influence it [16,19]. The aim of this study was twofold: first, to explore how attentional control is modulated when salience and value compete with one another for selection (Experiment 1); and second, how this selection is constrained when value becomes an integral part of the goal (Experiment 2). In Experiment 1, threat conveyed by a fearful face was used to create a bottom-up guidance of attention, besides luminance known to facilitate exogenous attention as a low-level perceptual cue. More specifically, in Experiment 1, we used a DPT [45] during which a low-level visual change in terms of luminance at the cue level occurred either at the same or a different location compared to the fearful (or happy) face. We hypothesized that if threat and luminance both correspond to salient events and hence they each engage exogenous attention, then an additive effect for them could be found whereby target processing would depend on their linear combination. In Experiment 2, we used a similar DPT but altered cue-target probability based on the emotional face to assess if participants could use this information or knowledge at the cue level to steer endogenous attention and hence facilitate target processing depending on it. Hence, we borrowed the logic of attention bias modification [64]. We hypothesized that if threat corresponds to a genuine salient event, then target processing should not be influenced by cue-target probability but mostly this exogenous cue instead. However, if threat processing depends on endogenous attention control mechanisms, then a larger capture by fearful faces should be observed in the high than low probability condition. A number of important new results emerge from these two complementing experiments, as discussed here below.

First and foremost, the results of Experiment 1 clearly show response facilitations for valid compared to invalid trials when validity was defined by exogenous attention (i.e., Validity-E). This result unambiguously shows that the low-level

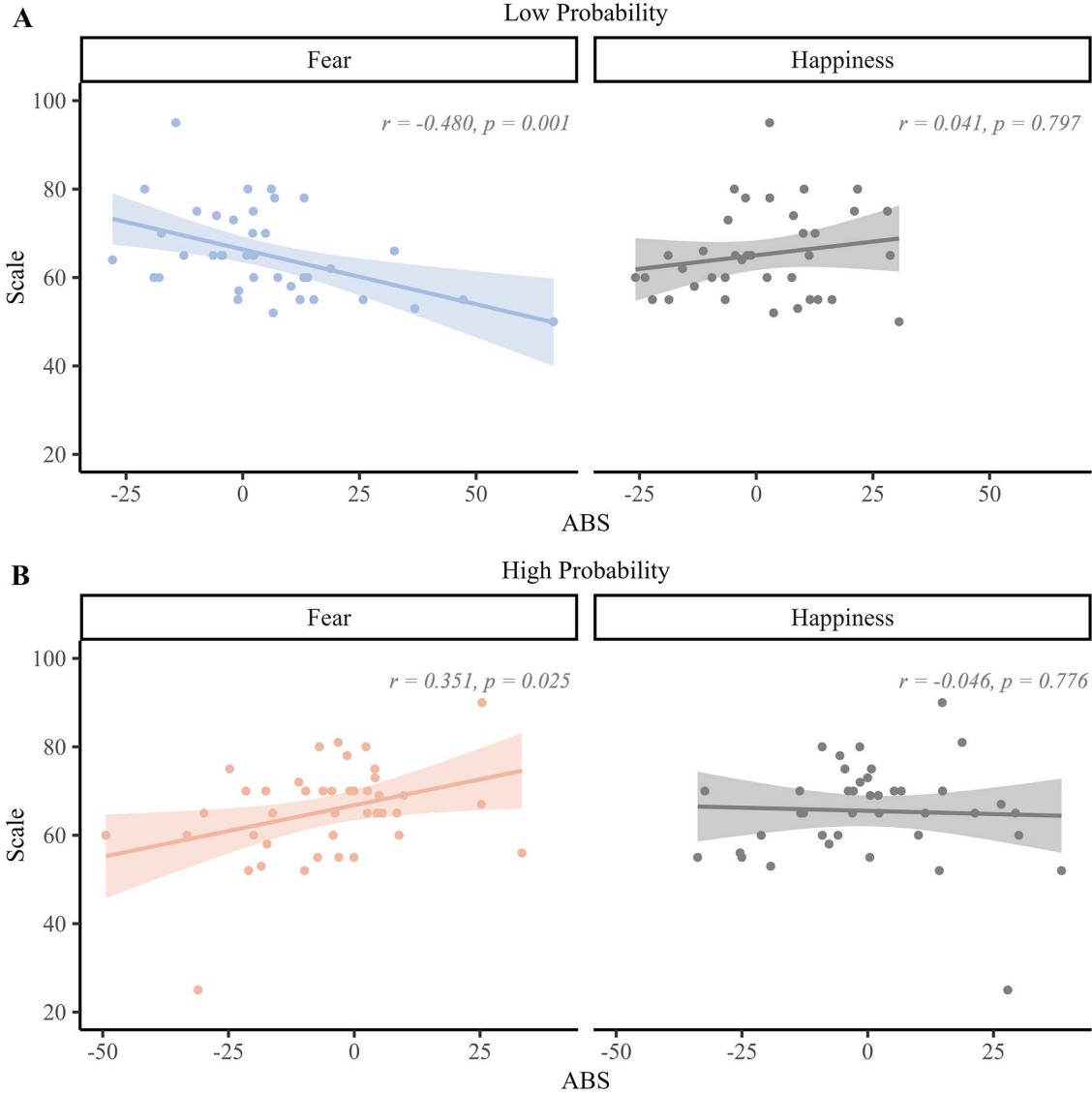

**Fig 5. Correlations between awareness and the ABS (based on RTs), separately for each emotion and probability.** Each correlation is shown using a scatterplot with the regression line along with the 95% confidence interval (CI).

**Table 6. Results of the GLM used in Experiment 2 to compare fear and happiness for their propensity to influence awareness.**

|  |  | Estimate | Std. Error | T value | Pr(>\|t\|) |
|---|---|---|---|---|---|
| Low probability | (Intercept) | 66.720 | 1.450 | 46.009 | < 0.001 |
|  | F_ABSes | −0.242 | 0.071 | −3.409 | 0.002 |
|  | H_ABSes | 0.034 | 0.053 | 0.643 | 0.524 |
| High probability | (Intercept) | 66.816 | 1.720 | 38.850 | < 0.001 |
|  | F_ABSes | 0.233 | 0.101 | 2.299 | 0.027 |
|  | H_ABSes | −0.020 | 0.100 | −0.212 | 0.833 |

*Note*: F_ABSes: attentional bias score for fear; H_ABSes: attentional bias score for happiness.

visual change based on luminance captured attention exogenously [8,9,46]. Previous studies have already shown that abrupt luminance onsets [4,11] and sudden changes in luminance [65,66] can capture attention in a stimulus-driven fashion, indicating that luminance corresponds to a genuine low-level visual property that influences exogenous attention. It has been proposed that abrupt visual events capture attention by activating the magnocellular pathway, which is sensitive to rapid and transient visual changes [67,68]. Because the main effect of Validity-E was significant and this factor did not interact with Emotion, our results (Experiment 1) suggest that the abrupt and transient luminance change occurring at the cue level facilitated target processing at this location equally strongly for fearful and happy faces [15,30,47]. Orthogonally to this exogenous attention effect, we also found in Experiment 1 a significant interaction between Emotion and Validity-V, replicating our previous results [45] and showing that negative valence, as opposed to threat per se, likely captured attention in this DPT [17,21,69]. Accordingly, participants were faster for happy invalid than happy valid trials (while the symmetric facilitation effect for fear valid compared to fear invalid trials was weaker and not significant); an interaction effect between Emotion and Validity-V that we already previously reported and suggesting that negative emotion (but not threat per se) captured spatial attention in this DPT [45]. Even though we did not ask the participants to rate the different faces used in our experiment (mostly due to time constraints) and hence we could not directly confirm that neutral faces were actually perceived as negative stimuli, based on independent results obtained using a similar DPT and the same emotional faces [45,70], we have however good reasons to assume that it was indeed the case. In these previous studies [45,70], performance for the induction trials revealed that it was more challenging for the participants to discriminate neutral from fearful faces than neutral from happy faces, as if a shorter distance on a putative valence axis or continuum was used in the former compared to the latter condition [71]. As a result of this asymmetry between fearful and happy faces (relative to neutral faces), the competition for attentional selection was likely increased during the DPT between the former ones and neutral faces, eventually leading to comparable RTs for fear-valid and fear-invalid trials. However, when the same neutral faces competed for attention selection with happy faces, then attentional selection was likely biased more easily or swiftly towards the former emotion category because they could signal negativity or threat [72,73]. This in turn led to an enhanced invalidity effect for happy faces. Hence, besides luminance, negative emotion was used as an exogenous cue by the participants to steer attentional control in this DPT. Because of this invalidity effect, one could argue that Inhibition of Return (IOR) could drive this Validity-V effect. IOR refers to the inhibition of a previously explored location to facilitate the orienting towards novel ones [74]. This is typically expressed by slower RTs for valid than invalid trials when a long SOA (i.e., above 300 ms) between cue and target is used. To test this alternative account, we ran another analysis where we re-analyzed the data of Experiment 1 using SOA as an additional factor. In this control analysis, the main effect of SOA was significant (showing faster RTs for trials with a long than short SOA), but importantly, the main effect of Validity-E and interaction effects (Emotion × Validity-V, Validity-V × Validity-E) remained significant, regardless of SOA. Hence, the results of this control analysis ruled out an interpretation of these results in terms of IOR.

However and crucially, these two exogenous effects, luminance and negative emotion/valence, did not add together to determine attentional control. Instead, we found a significant interaction between Validity-E and Validity-V. Specifically, we found that the luminance's change captured attention when emotion was also valid. If emotion was invalid (i.e., the emotional cue and the luminance cue were shown on two competing locations along the horizontal axis), then the capture of spatial attention by luminance was not found. Accordingly, negative emotion interfered with luminance, suggesting that these two cues were not used in an additive manner to determine exogenous attention [47]. Instead, these two factors appeared to interact in a manner consistent with the under-additivity hypothesis [48] according to which their combined effect did not equal the sum of their independent effects. In other words, when both cues were valid, one of them, probably luminance because it was likely more salient than value, was prioritized by (exogenous) attention control mechanisms. However, if negative emotion was not spatially compatible with luminance, then the capture by the latter component decreased because "another" salient cue (i.e., emotion) was shown at a different and spatially incompatible location. In light of these results and more specifically the under-additivity pattern found in Experiment 1, we reckon that negative

value is not simply equivalent to luminance in terms of salience, but these two "exogenous" cues appear to yield competition or mutual inhibition between them during bottom-up attention selection (see Fig 6A). Hence, the results of Experiment 1 suggest that negative valence can be conceived as a salient cue to some degree and thereby it can influence exogenous attention [22,23], yet it seems to compete with another salient cue defined by luminance during attention selection, suggesting a possible distinction between physical and emotional salience [21,75].

In contrast with these results, the findings of Experiment 2 where endogenous attention was promoted by means of a standard cue-target probability manipulation [15], neither the interaction between Emotion and validity, nor Probability (or the interaction between this endogenous attention factor and the two other ones) reached significance. Hence, the results of Experiment 2 did not reveal a capture of spatial attention by negative emotion (as found in Experiment 1 or in our previous study, see [45]) and/or a guidance of it based on the knowledge provided to the participants about the cue-target association compatible with endogenous attention. We surmise that attentional control could have been somewhat disrupted in Experiment 2 because goal-directed attention could not be easily activated or used based on emotion and/or fearful and happy faces did not provide clear visual stimuli or information to bias spatial attention. Moreover, because we used a short and variable cue-target interval, it is possible that the participants could not use the emotional information used at the cue level to bias in a top-down manner target processing accordingly. However and importantly, the results of Experiment 2 also showed that awareness of this association actually determined endogenous attention based on threat, selectively (see also [76]). More specifically, in the low-probability condition where the target mostly replaced the position of the neutral face in the pair used at the cue level, the attentional bias for fear was inversely related to the awareness of this association while in the high-probability condition where the target mostly appeared on the same side as the emotional face, this relationship between fear and awareness swapped and became positive. Importantly, these results only held for fear, but not happiness. In other words, these results showed that participants who were aware of the cue-target association could use it somehow during the DPT to steer attentional control in a top-down manner, yet if and only if a fearful face was shown at the cue level. These results are intriguing and they align with recent findings showing that the capture of spatial attention by fearful faces is deemed top-down because it depends on awareness and is modulated by task demands [77,78]. Although attention and awareness can be dissociated from each other [79,80], the results of

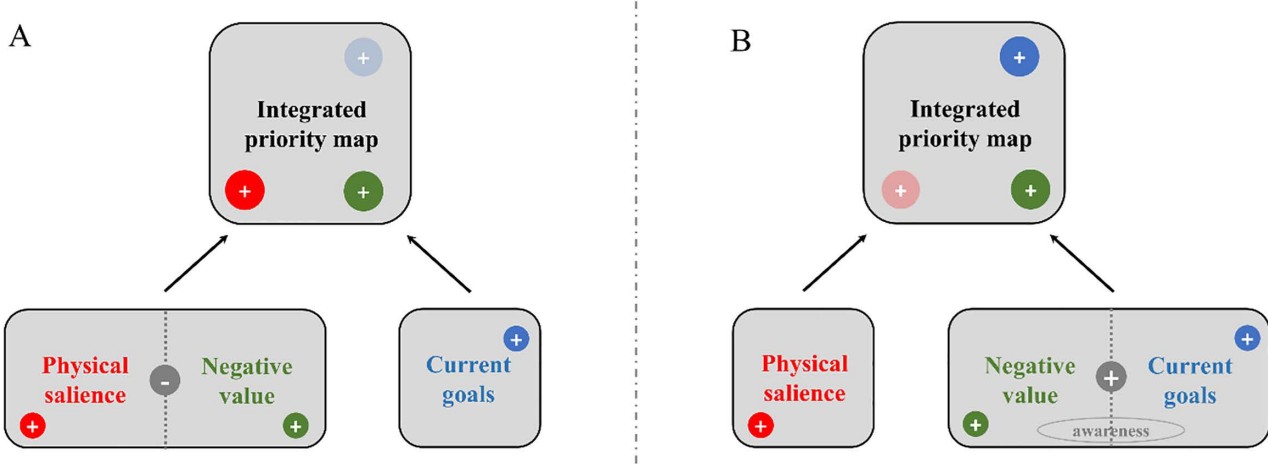

**Fig 6. Revised priority map (figure adapted based on Awh et al. [16]).** (A) Value (here corresponding to negative valence mostly) and physical salience (here with luminance as a main distinctive low-level visual property) can each contribute to bias spatial attention in a bottom-up or exogenous way. Yet, they can mutually inhibit each other during attention selection and eventually lead to under-additivity. (B) Fear, unlike happiness, can bias top-down or endogenous attention, yet this effect is mediated by awareness of the cue-target association.

Experiment 2 converge with theoretical accounts assuming that endogenous attention can actually be fostered by aware-ness [15,81–83]. Hence, even though phenomena such as "attention without awareness" [84,85] or "awareness without attention" [86] have been reported in the past, here we show that spatial attention is biased by threat-related stimuli (i.e., fearful faces) when participants were aware of the association created between (negative) emotion and target's location. This result accords with previous studies using binocular rivalry and continuous flash suppression (CFS), and showing that fearful faces can gain preferential access to awareness compared to neutral or positive faces [87–90]. Furthermore, as previously shown by Pessoa et al. [91] using backward masking and neuroimaging, large inter-individual differences are usually observed for this attentional bias. Likewise, here we show that awareness of the cue-target association varied substantially across participants of Experiment 2 (Fig 5) but intriguingly, for those who were aware of it, fear but not happi-ness could eventually be used to guide and facilitate target processing. Accordingly, these findings suggest that fear could bolster endogenous attention in this DPT because this emotion category could more easily break into awareness than neutral or positive emotions [82,83,91]. As depicted in Fig 6B, we therefore surmise that negative value could assist goal to determine the priority map and hence influence attentional control mostly because negative value promotes awareness.

## Limitations

A few limitations warrant comment. First, for practical reasons, we could not assess effects of negative value on either exogenous or endogenous attention using a fully within-subject design. Hence, we tested different participants in Exper-iments 1 and 2. Because they were tested at the same time and were recruited from the same student population and using the same online platform (Sona), it is unlikely that uncontrolled group differences accounted for these results. However, because the STAI results showed that anxiety levels were unexpectedly higher in Experiment 1 than Exper-iment 2, we believe that to get a comprehensive understanding of the interplay between exogenous, endogenous and emotional attention, it appears important in future studies to explore their combination using a within-subject design [47]. Another limitation concerns the reliance on self-reports exclusively to measure awareness in Experiment 2. Although the VAS administered after each session provided a straightforward means to probe participants' awareness of the cue-target association, it might be susceptible to biases or expectations and not suited to capture trial by trial fluctuations of awareness over the course of the experiment. To address this limitation, in future studies, more frequent and objective measures of awareness could be harnessed [92,93], as well as confidence ratings [94]. Third, gender was unbalanced in our experiments, with a majority of female participants included in our samples, thereby limiting the possibility to explore the role of this variable on emotional attention. Although a control analysis (reported in the S1_file) revealed that this factor, when included as a covariate in the statistical analyses, did not lead to a main effect or significant interactions with the other factors, emotional attentional effects (for Experiment 2) turned out to be attenuated in this analysis, mostly due to this imbalance. Thus, it remains to be shown in the future whether the complex interaction effects found between salience, goal and value in this study could hold for male and female participants equally strongly, or gender differences might emerge.

## Conclusions

The results of this study provide new insights into the interplay between exogenous, endogenous and emotional factors during attentional control. When directly competing with a low-level and neutral visual feature (i.e., luminance, Experiment 1), negative value, here conveyed by a facial expression, can also be used by the participants to unlock an exogenous and stimulus-driven control of attention. Yet, emotional salience does not simply add up to physical salience in this situa-tion, but they interact with each other, probably to maximize salience processing when it is determined by multiple cues. Interestingly, the same cue (i.e., negative value) can also be used to guide goal-directed attention, yet if and only if partici-pants are aware of the association created between emotion and target's location (Experiment 2), thereby also confirming the existence of close ties between endogenous attention and awareness. All in all, these results underscore the flexibility

of negative value and its propensity to influence attentional control either in a seemingly bottom-up or top-down manner depending on the specific configuration of cues available in the environment to foster this process.

## Supporting information

**S1_File. Supplementary information including additional analyses.**
(PDF)

## Author contributions

**Conceptualization:** Xiaojuan Xue, Gilles Pourtois.

**Data curation:** Xiaojuan Xue.

**Formal analysis:** Xiaojuan Xue.

**Funding acquisition:** Xiaojuan Xue.

**Investigation:** Gilles Pourtois.

**Methodology:** Xiaojuan Xue, Gilles Pourtois.

**Project administration:** Xiaojuan Xue.

**Resources:** Xiaojuan Xue, Gilles Pourtois.

**Software:** Xiaojuan Xue.

**Supervision:** Xiaojuan Xue, Gilles Pourtois.

**Validation:** Xiaojuan Xue.

**Visualization:** Xiaojuan Xue.

**Writing – original draft:** Xiaojuan Xue, Gilles Pourtois.

**Writing – review & editing:** Xiaojuan Xue, Gilles Pourtois.

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
