## [Decision Letter · Decision Letter 0]

18 Jul 2025

Dear Dr. Xue,

Thank you for submitting your manuscript to PLOS ONE. After careful consideration, we feel that it has merit but does not fully meet PLOS ONE’s publication criteria as it currently stands. Therefore, we invite you to submit a revised version of the manuscript that addresses the points raised during the review process.

We look forward to receiving your revised manuscript.

Kind regards,

Giulia Prete

Academic Editor

PLOS ONE

Journal Requirements:

“This work was supported by a grant (JY202126) from the Guangzhou Elite Scholarship Council (GESC) awarded to Xiaojuan Xue.”

Please provide an amended statement that declares *all* the funding or sources of support (whether external or internal to your organization) received during this study, as detailed online in our guide for authors at http://journals.plos.org/plosone/s/submit-now . Please also include the statement “There was no additional external funding received for this study.” in your updated Funding Statement.

“This work was supported by a grant (JY202126) from the Guangzhou Elite Scholarship Council (GESC) awarded to Xiaojuan Xue.”

5. We note that Figure 1 includes an image of a participant in the study.

Reviewers' comments:

Reviewer's Responses to Questions

**Comments to the Author**

1. Is the manuscript technically sound, and do the data support the conclusions?

Reviewer #1: Yes

Reviewer #2: Partly

2. Has the statistical analysis been performed appropriately and rigorously?

Reviewer #1: Yes

Reviewer #2: Yes

3. Have the authors made all data underlying the findings in their manuscript fully available?

Reviewer #1: Yes

Reviewer #2: No

4. Is the manuscript presented in an intelligible fashion and written in standard English?

Reviewer #1: Yes

Reviewer #2: No

Reviewer #1: Comments to the authors:

This manuscript targets an equally important and interesting research topic. It is very well written, concise, comprehensible and transparent. It presents a sound method and offers new insights into the cross-over field of attentional and emotional processing. Therefore, I have only very view minor suggestions for improvement. Inclusion of most of those, in my opinion, do not necessarily require another review but solely a close read-through by the editor. I would suggest another review round, if new analyses and interpretation would be required, please see my comment on gender distribution. Please find attached my few comments:

- Abstract: “target’s location at the block level” – without the exhaustive description of the paradigm - you nevertheless offer at a later point – it could be difficult for readers to know what “block level” means. Consider rephrasing it using e.g. condition (like yourself used later in the manuscript) or similar.

- Introduction: page 4, line 61 – “(2) value is an integral part…”, consider putting “whether” before “value” for completion of the research question

- Introduction: page 6, paragraph starting with line 124 – please consider explaining your previous study in more detail for better comprehension of itself and the topic of the current manuscript.

- Introduction: page 6, line 127 – please whether a typo occurred: “but also symmetrically for happy invalid than happy invalid trials”. Furthermore, without further description of the study itself, I do not fully understand, what “symmetrically” means in this context

- Section “Participants” – Your (otherwise sound) samples present an unbalanced gender ratio (17% and 19%, respectively). I would suggest you include the (preferably absent) effects of gender on the investigated specific attentional and emotional processes in the introduction section. If gender effects cannot be ruled out, consider including it as a covariate in your analyses.

- Procedure: page 11, paragraph starting with line 238 – Please consider putting the information regarding the STAI into the participants section

- Discussion: Page 19, line 418 – Please explain / elaborate the construct of IOR to the readers

Reviewer #2: This manuscript presents a potentially valuable contribution by proposing a novel priority map to explain spatial attention biases involving both exogenous and endogenous factors. The research question is timely and thought-provoking, and several aspects of the manuscript are well-articulated. However, there are important areas where greater clarity and theoretical integration are needed to enhance the manuscript’s rigor and replicability. In particular, the comparison between Experiments 1 and 2 would benefit from a more thorough theoretical and methodological justification, especially in light of the change in target type (square vs. dot). This shift introduces potential perceptual confounds that may influence reaction times independently of attentional bias. Addressing this limitation explicitly would strengthen the interpretation of the findings and support more valid cross-experiment discussions (as is central to the authors’ proposed revision of the priority map formulated in Figure 5). Some factors are inconsistently labeled and insufficiently explained across the Results and Discussion sections, which complicates interpretation and may obscure key effects. Additionally, while the inclusion of the STAI is potentially relevant, its theoretical role in the current study remains underdeveloped and would benefit from clearer motivation and discussion. Overall, with careful attention to the issues noted—particularly regarding experimental design, factor labeling, and theoretical framing—the manuscript has the potential to make a meaningful contribution, particularly if revised for a more specialist publication outlet.

Introduction

p.4: “For example, fear (conveyed by a fearful face) can enhance contrast sensitivity (24) and this perceptual effect appears to be mostly mediated by the processing of coarse/low-spatial frequency information (25).” Similar claims have also been made with respect to the facilitative effects of processing happy facial expressions:

Calvo, M. G. and Marrero, H. (2008). Visual search of emotional faces: The role of a�ective content and featural distinctiveness. Cognition and Emotion 22, 1-25.

Calvo, M. G. and Nummenmaa, L. (2008). Detection of Emotional Faces: Salient physical features guide e�ective visual search. Journal of Experimental Psychology: General 137, 471-494.

The authors are encouraged to review this work and incorporate it alongside the arguments presented on page 4 in order to add a more nuanced perspective on the topic.

p.4: It would be helpful for the authors to specify here the tasks used to assess faster/better processing of threat-related stimuli in the following statement: “as revealed by faster/better processing for them compared to neutral or even positive stimuli (26–28).”

p.5: DPT abbreviation should follow “dot-probe task”.

p.5: The authors should clarify whether low-level physical change was manipulated for happy faces as well in the following statement: “As a control condition, we used happy

faces.”

p.5: The authors need to describe in more detail how induction trials were used to activate a specific goal in their description of the following: “For example, Vogt et al. (44) used a DPT in combination with induction trials meant to activate a specific goal and found that negative stimuli only captured attention when they were directly task-relevant and hence goal-relevant (40,44,48– 50).”

p.6: The authors should clarify how the low vs. high probability conditions are defined, as the current explanation is confusing. The sentence suggests that in both conditions the target appears on the same side as a specific face type (neutral or emotional) in 75% of trials, which makes it unclear what distinguishes the two conditions. Please specify more clearly how the probability differs between conditions and which face type serves as the probabilistic cue in each case. “In agreement with these instructions, in the low probability condition, the target appeared on 75% of the trials on the same side as the neutral face while in the high probability condition, it appeared on 75% of the trials on the same side as the emotional face instead.”

Methods

p.7: What is ACC? It is the first time this abbreviation is used.

p.8: The authors should specify the Ekman and Friesen dataset here instead of “Ekman dataset.”

p.8: Possible typo: the tick one/the thick one.

p.9: The authors need to explain why were different SOAs used. The analysis by SOA should be included in a dedicated sub-section of the Results, rather than the supplemental section in order to facilitate the interpretation of the results on pages 19-20 in the Discussion.

p.9: The level of detail regarding the instructions provided to participants is insufficient: “Participants were asked to discriminate the shape of the target, either a diamond or a square, as accurately and quickly as possible. Speed was emphasized and it provided the main dependent variable in both experiments.” Please provide full details regarding what the participants were required to do when they saw a face on each trial, how was speed emphasised to participants and whether any feedback provided.

p.10: When Experiment 2 is introduced on page 10, the authors should clarify whether luminance was also manipulated.

p.10: “a visual analog scale (VAS). This VAS consisted of a continuous scale ranging from 0 (no awareness of the association) to 100 (full awareness of the association). Hence, higher scores on this VAS indicated greater awareness at the subjective level. At the beginning of each session, the participants received specific instructions (on screen).” It is not clear what type of awareness the VAS measures. If participants received instructions about the probability of a target appearing after an emotional or neutral face at the beginning of each block, what exactly were the participants asked to reflect on in the VAS?

p.10: The authors should clarify if the SOAs used in Experiment 2 were equiprobable (as described in Experiment 1).

p.10: “the target (either a dot or a square) was shown for 150 ms at the location previously occupied by the emotional face (valid trial) or the neutral face (invalid trial).” Why was the target changed from diamond to dot in Experiment 2?

p.10: Inconsistency with target description here: “diamond and their right index finger for the square”, when in the previous sentence the target is described as a dot.

p.11: “At the end of the experiment, participants were asked to complete the trait version of the State-Trait Anxiety Inventory (STAI-T; see Spielberger (55)). We used the twenty-item version (Form Y-2) that assessed trait anxiety. All items were rated on a 4-point scale (e.g., from “Almost Never” to “Almost Always”). Scores range from 20 to 80, with higher scores reflecting higher levels of anxiety. The average score of the STAI was 43.80 (range 26-68; SD: 10.265) in Experiment 1 and 38.27 (range 29-63; SD: 8.155) in Experiment 2. The mean score in Experiment 2 was significantly lower than in Experiment 1 ( 78 = -2.673, p = 0.009, Cohen’s d = -0.598). Despite this significant difference, both scores were lower than the commonly used cutoff of 45 for clinical anxiety, indicating that neither group reached clinical anxiety.” What is the theoretical motivation for administering the STAI? Why is a cut-off of 45 used (and where is the citation to support it?). The authors are encouraged to consult the STAI manual for relevant associated norms and conduct relevant comparison tests with these values.

Spielberger, C. D., Gorsuch, R. L., Lushene, R., Vagg, P., & Jacobs, G. (1983). Manual for the state-trait anxiety inventory. Consulting Psychologists Press.

Data Analysis

p. 11-12: It’s not clear what the Total and Correct trials column refers to in Tables 1 and 2.

p.13: The way the factors are labelled is a little confusing: In the Data Analysis section on page 12 one of the factors is presented as: “Validity-V (here “V” refers to value of the face: the target replaced either the emotional or neutral face at the cue level)”, but in the Results section of Experiment 1 on page 13 it is described as “ anecdotal evidence for including this effect), showing faster RTs for Invalid than Valid trials (see Fig 2B).”

p.13: Could the following be contextualised a little better: “with faster RTs for happy than fearful trials.” Does this perhaps refer to faster response times to the target when it is preceded by happy face trials than fearful trials”?

Discussion

p.17: What does F_ABS refer to?

p.18: The labelling of factors is confusing and difficult to follow throughout the discussion. The authors are encouraged to come up with a more meaningful way to label the factors than Validity E.

p.19: The authors should clarify whether the luminance change affected the processing of the target equally for happy and fearful faces in the following statement: “luminance change occurring at the cue level was processed as a salient event by the participant, thereby facilitating target processing when it was shown at the same location as this change (15,28,42).”

p.19: The authors claim that “neutral faces are perceived as negative stimuli in this task,” How have the authors established whether this is the case? Were the neutral faces rated negatively by participants? Where is the data to support this assertion?

References

p.28: Author order correction: Ekman, P, Friesen, W.V.

**Do you want your identity to be public for this peer review?** For information about this choice, including consent withdrawal, please see our Privacy Policy

Reviewer #1: No

Reviewer #2: No

---

## [Author Response · Author response to Decision Letter 1]

27 Aug 2025

Reviewer #1: Comments to the authors:

This manuscript targets an equally important and interesting research topic. It is very well written, concise, comprehensible and transparent. It presents a sound method and offers new insights into the cross-over field of attentional and emotional processing. Therefore, I have only very view minor suggestions for improvement. Inclusion of most of those, in my opinion, do not necessarily require another review but solely a close read-through by the editor. I would suggest another review round, if new analyses and interpretation would be required, please see my comment on gender distribution. Please find attached my few comments:

Response: We thank the reviewer for their positive evaluation of our work and manuscript. We have revised the manuscript according to their valuable comments.

1. - Abstract: “target’s location at the block level” – without the exhaustive description of the paradigm - you nevertheless offer at a later point – it could be difficult for readers to know what “block level” means. Consider rephrasing it using e.g. condition (like yourself used later in the manuscript) or similar.

Response: Thank you for this suggestion. We have rephrased this sentence.

Original:

“fear, unlike happiness, could bias spatial attention in a top-down manner, yet only when participants were aware of the association created between the emotional cue and target’s location at the block level. ”

Revised:

“fear, unlike happiness, could bias spatial attention in a top‑down manner, but only when participants were aware of the association created between the emotional cue and target’s location, leading to an enhanced validity effect in the high probability condition but invalidity effect in the low probability one.”

2. - Introduction: page 4, line 61 – “(2) value is an integral part…”, consider putting “whether” before “value” for completion of the research question

Response: Thank you for this suggestion. However, to maintain the parallelism between these two conditions, we have revised the sentence using “when…when” rather than “whether.”

Original:

However, two unanswered questions are how is attentional control achieved when (1) salience and value directly compete with one another for selection, and (2) value is an integral part or component of the goal.

Revised:

“However, two unanswered questions are how is attentional control achieved (1) when salience and value directly compete with one another for selection, and (2) when value is an integral part or component of the goal. ”

3. - Introduction: page 6, paragraph starting with line 124 – please consider explaining your previous study in more detail for better comprehension of itself and the topic of the current manuscript.

Response: Thanks for your suggestion. We have substantially expanded the description of our previous study to provide readers with a clearer context:

Original:

In this previous study, we found that in the absence of induction trials (i.e. baseline DPT), participants were faster for fear valid than fear invalid trials, but also symmetrically for happy invalid than happy invalid trials (even though emotion was never predictive of target’s location), suggesting that negative emotion, as opposed to fear or threat per se, actually captured attention in this task.

Revised:

“In this previous study, we compared attentional biases to emotional faces using a DPT alone where emotion was never goal-relevant (Experiment 1) or made directly task-relevant by means of induction trials (Experiments 2-3). We found out that, in the absence of induction trials (Experiment 1), negative faces captured attention, with faster target processing when it appeared on the same side as the preceding fearful face (i.e., fear-valid trials) compared to the opposite side where the neutral face was shown (i.e., fear-invalid trials), but also when it appeared on the side of the preceding neutral face (i.e., happy-invalid trials) compared to the happy face (i.e., happy-valid trials), suggesting that negative emotion, as opposed to fear or threat per se, actually captured attention in this task”

4. - Introduction: page 6, line 127 – please whether a typo occurred: “but also symmetrically for happy invalid than happy invalid trials”. Furthermore, without further description of the study itself, I do not fully understand, what “symmetrically” means in this context

Response: Thanks for pointing that out. See our response to point #3 raised here above. We have rewritten that paragraph. We believe this revised wording addresses both your request for more detail on our previous study as well as the typo/unclear use of the adverb “symmetrically.”

5. - Section “Participants” – Your (otherwise sound) samples present an unbalanced gender ratio (17% and 19%, respectively). I would suggest you include the (preferably absent) effects of gender on the investigated specific attentional and emotional processes in the introduction section. If gender effects cannot be ruled out, consider including it as a covariate in your analyses.

Response: We appreciate the reviewer’s suggestion regarding the potential influence of gender. To address this concern, we re-ran our statistical analyses including Gender as a covariate (we added these results in the supplementary material section). While some of the originally significant effects decreased and became marginally significant, the overall pattern of results remained, however. This attenuation is likely due to the inclusion of Gender as an additional factor, which is imbalanced and reduces statistical power, rather than indicating a systematic modulation of emotional attention depending on it. More importantly, the results consistently show no significant main effect of Ggender and no significant interactions between Ggender and the other experimental factors, indicating that Ggender did not influence systematically the observed attentional effects. Given these findings, we interpret the main results with caution, however, and directly acknowledge the unbalanced gender ratio as a potential limitation of our study in the revised manuscript on p.27.

6. - Procedure: page 11, paragraph starting with line 238 – Please consider putting the information regarding the STAI into the participants section

Response: As suggested, we have moved the details regarding the STAI from the Procedure section to the Participants section.

7. - Discussion: Page 19, line 418 – Please explain / elaborate the construct of IOR to the readers

Response: Thank you for your suggestion. We have added a brief description of IOR so that readers unfamiliar with this term and attentional effect can follow our argument.

Original:

“Because of this invalidity effect, one could argue that Inhibition of Return (IOR) could drive this Validity-V effect.”

Rivised:

“Because of this invalidity effect, one could argue that Inhibition of Return (IOR) could drive this Validity-V effect. IOR refers to the inhibition of a previously explored location to facilitate the orienting towards novel locations [1]. This is typically expressed by slower RTs for valid than invalid trials when a long SOA (i.e., above 300 ms) between cue and target is used.”

Reviewer #2: This manuscript presents a potentially valuable contribution by proposing a novel priority map to explain spatial attention biases involving both exogenous and endogenous factors. The research question is timely and thought-provoking, and several aspects of the manuscript are well-articulated. However, there are important areas where greater clarity and theoretical integration are needed to enhance the manuscript’s rigor and replicability. In particular, the comparison between Experiments 1 and 2 would benefit from a more thorough theoretical and methodological justification, especially in light of the change in target type (square vs. dot). This shift introduces potential perceptual confounds that may influence reaction times independently of attentional bias. Addressing this limitation explicitly would strengthen the interpretation of the findings and support more valid cross-experiment discussions (as is central to the authors’ proposed revision of the priority map formulated in Figure 5). Some factors are inconsistently labeled and insufficiently explained across the Results and Discussion sections, which complicates interpretation and may obscure key effects. Additionally, while the inclusion of the STAI is potentially relevant, its theoretical role in the current study remains underdeveloped and would benefit from clearer motivation and discussion. Overall, with careful attention to the issues noted—particularly regarding experimental design, factor labeling, and theoretical framing—the manuscript has the potential to make a meaningful contribution, particularly if revised for a more specialist publication outlet.

Response: We thank the reviewer for their positive evaluation of our manuscript, as well as the thoughtful suggestions made to improve it further. We performed a point-to-point revision of it according to them (see here below for details).

Introduction

1. p.4: “For example, fear (conveyed by a fearful face) can enhance contrast sensitivity (24) and this perceptual effect appears to be mostly mediated by the processing of coarse/low-spatial frequency information (25).” Similar claims have also been made with respect to the facilitative effects of processing happy facial expressions:

Calvo, M. G. and Marrero, H. (2008). Visual search of emotional faces: The role of a�ective content and featural distinctiveness. Cognition and Emotion 22, 1-25.

Calvo, M. G. and Nummenmaa, L. (2008). Detection of Emotional Faces: Salient physical features guide e�ective visual search. Journal of Experimental Psychology: General 137, 471-494.

The authors are encouraged to review this work and incorporate it alongside the arguments presented on page 4 in order to add a more nuanced perspective on the topic.

Response: Thanks for your these thoughtful suggestions. We have revised the manuscript to convey a more nuanced discussion of this important topic and now directly refer to these earlier studies suggested by the reviewer. The revised text reads as follows:

“For example, fear (conveyed by a fearful face) can enhance contrast sensitivity (24) and this perceptual effect appears to be mostly mediated by the processing of coarse/low-spatial frequency information (25). However, happy faces, for which the upward-curved mouth provides an important diagnostic (low-level) feature, can also facilitate visual processing and bias spatial attention compared to neutral faces (26,27). Hence, although threat-related stimuli can be prioritized during attentional selection, probably due to their enhanced evolutionary relevance (28–30), positive stimuli can also bias spatial attention under specific circumstances.”

2. p.4: It would be helpful for the authors to specify here the tasks used to assess faster/better processing of threat-related stimuli in the following statement: “as revealed by faster/better processing for them compared to neutral or even positive stimuli (26–28).”

Response: Thanks for your this suggestion. In the revised manuscript, we now specify the experimental tasks that have demonstrated faster threat detection.

Original:

These results align with many other ones showing that threat-related stimuli capture attention involuntarily, as revealed by faster/better processing for them compared to neutral or even positive stimuli [2–4].

Revised:

These results align with many other ones showing that threat-related stimuli capture attention involuntarily, as revealed by faster/better processing for them compared to neutral or even positive stimuli in various tasks and contexts [2–4], including visual search [3,5], attentional blink [6,7], cueing [8,9], or dot-probe task (DPT) [10,11].

3. p.5: DPT abbreviation should follow “dot-probe task”.

Response: As suggested, we have added it.

4. p.5: The authors should clarify whether low-level physical change was manipulated for happy faces as well in the following statement: “As a control condition, we used happy

faces.”

Response: We clarified this issue and confirmed it was indeed the case. In Experiment 1, the low-level change in terms of luminance was exactly the same for fearful and happy faces. We have added the following sentence to make this explicit:

“As a control condition, we used happy faces. In this condition, the same abrupt luminance change occurred on one side of the display, as was done with fearful faces, eventually yielding a comparable exogenous cueing for these two emotion categories.”

5. p.5: The authors need to describe in more detail how induction trials were used to activate a specific goal in their description of the following: “For example, Vogt et al. (44) used a DPT in combination with induction trials meant to activate a specific goal and found that negative stimuli only captured attention when they were directly task-relevant and hence goal-relevant (40,44,48– 50).”

Response: Thank you for your this suggestion. In the revised manuscript (page 5), we have clarified this point:

“For example, Vogt et al. (44) used a DPT in combination with induction trials meant to activate a specific goal and found that negative stimuli only captured attention when they were directly task-relevant and hence goal-relevant (40,44,48–50). More specifically, during the induction trials, the participants were asked to press a key whenever a specific and pre-defined (emotional) stimulus was presented but to withhold responding to other stimuli, making this former stimulus task and hence goal relevant.”

6. p.6: The authors should clarify how the low vs. high probability conditions are defined, as the current explanation is confusing. The sentence suggests that in both conditions the target appears on the same side as a specific face type (neutral or emotional) in 75% of trials, which makes it unclear what distinguishes the two conditions. Please specify more clearly how the probability differs between conditions and which face type serves as the probabilistic cue in each case. “In agreement with these instructions, in the low probability condition, the target appeared on 75% of the trials on the same side as the neutral face while in the high probability condition, it appeared on 75% of the trials on the same side as the emotional face instead.”

Response: Thank you for your this suggestion. We have revised the manuscript on page 6 to clarify this issue. Low vs. High probability simply refers to the likelihood (either low or high) that the target appeared on the same side as the emotional face at the cue level. Accordingly, in the low probability condition, the target appeared on 75% of the trials on the same side as the neutral face, while it was shown on 25% of the trials on the same side as emotional face instead. In comparison, in the high probability condition, this mapping was reversed: it appeared on 75% of the trials on the same side as the emotional face, whereas on 25% of the trials, it was shown on the same side as the neutral face.

Methods

7. p.7: What is ACC? It is the first time this abbreviation is used.

Response: We apologize for this. ACC simply means Accuracy and we have abbreviated it upon its first occurrence in the text on p.8.

8. p.8: The authors should specify the Ekman and Friesen dataset here instead of “Ekman dataset.”

Response: Thanks for pointing this out. We revised the wording accordingly.

9. p.8: Possible typo: the tick one/the thick one.

Response: Thank you for noticing this typo that we have corrected.

10. p.9: The authors need to explain why were different SOAs used. The analysis by SOA should be included in a dedicated sub-section of the Results, rather than the supplemental section in order to facilitate the interpretation of the results on pages 19-20 in the Discussion.

Response: We thank the reviewer for this comment and suggestion. As we now better explain on p. 11 in the revised manuscript, we mostly used these different/variable SOAs to prevent temporal attention effects and in keeping with Pourtois et al. (2004) where this variability was introduced (with the DPT). As a result of it

---

## [Decision Letter · Decision Letter 1]

11 Sep 2025

Neither exogenous, nor endogenous: evidence for a distinct role of negative emotion during attentional control

PONE-D-25-06615R1

Dear Dr. Xue,

We’re pleased to inform you that your manuscript has been judged scientifically suitable for publication and will be formally accepted for publication once it meets all outstanding technical requirements.

Kind regards,

Giulia Prete

Academic Editor

PLOS ONE

Additional Editor Comments (optional):

Please, consider to insert a new supplementary file containing raw data: The file labelled S1_File consists of data analysis rather than the underlying raw data. Thank you and good luck in your future research! 

Reviewer's Responses to Questions

**Comments to the Author**

Reviewer #1: All comments have been addressed

Reviewer #2: All comments have been addressed

2. Is the manuscript technically sound, and do the data support the conclusions?

Reviewer #1: Yes

Reviewer #2: Yes

3. Has the statistical analysis been performed appropriately and rigorously?

Reviewer #1: Yes

Reviewer #2: Yes

4. Have the authors made all data underlying the findings in their manuscript fully available?

Reviewer #1: Yes

Reviewer #2: No

5. Is the manuscript presented in an intelligible fashion and written in standard English?

Reviewer #1: Yes

Reviewer #2: Yes

Reviewer #1: You have adressed all my issues satisfyingly.

I recommend to accept this fine article.

Best regards

Reviewer #2: The authors have clearly and adequately addressed my comments raised in a previous round of review, but I was unable to locate the data underlying the findings described in their manuscript in the Supporting Information files. The file labelled S1_File consisted of analyses rather than the underlying data.

**Do you want your identity to be public for this peer review?** For information about this choice, including consent withdrawal, please see our Privacy Policy

Reviewer #1: No

Reviewer #2: No

---

## [Editor Report · Acceptance letter]

PONE-D-25-06615R1

PLOS ONE

Dear Dr. Xue,

I'm pleased to inform you that your manuscript has been deemed suitable for publication in PLOS ONE. Congratulations! Your manuscript is now being handed over to our production team.

Kind regards,

on behalf of

Dr. Giulia Prete

Academic Editor

PLOS ONE